| Open Peer Review | Biotechnology | Methods and Protocols

# A SEVA-based, CRISPR-Cas3-assisted genome engineering approach for *Pseudomonas* with efficient vector curing

Eveline-Marie Lammens,[1] Daniel Christophe Volke,[2] Kaat Schroven,[1] Marleen Voet,[1] Alison Kerremans,[1] Rob Lavigne,[1] Hanne Hendrix[1]

**ABSTRACT** The development of CRISPR-Cas-based engineering technologies has revolutionized the microbial biotechnology field. Over the years, the Class II Type II CRISPR-Cas9 system has become the gold standard for genome editing in many bacterial hosts. However, the Cas9 system does not allow efficient genomic integration in *Pseudomonas putida*, an emerging Synthetic Biology host, without the assistance of lambda-Red recombineering. In this work, we utilize the alternative Class I Type I-C CRISPR-Cas3 system from *Pseudomonas aeruginosa* as a highly efficient genome editing tool for *P. putida* and *P. aeruginosa*. This system consists of two vectors, one encoding the Cas genes, CRISPR array and targeting spacer, and a second Standard European Vector Architecture vector, containing the homologous repair template. Both vectors are Golden Gate compatible for rapid cloning and are available with multiple antibiotic markers, for application in various Gram-negative hosts and different designs. By employing this Cas3 system, we successfully integrated an 820-bp cassette in the genome of *P. putida* and performed several genomic deletions in *P. aeruginosa* within a week, with an efficiency of >83% for both hosts. Moreover, by introducing a universal self-targeting spacer, the Cas3 system rapidly cures all helper vectors, including itself, from the host strain in a matter of days. As such, this system constitutes a valuable engineering tool for *Pseudomonas*, to complement the existing range of Cas9-based editing methods, and facilitates genomic engineering efforts of this important genus.

**IMPORTANCE** The CRISPR-Cas3 editing system as presented here facilitates the creation of genomic alterations in *Pseudomonas putida* and *Pseudomonas aeruginosa* in a straightforward manner. By providing the Cas3 system as a vector set with Golden Gate compatibility and different antibiotic markers, as well as by employing the established Standard European Vector Architecture (SEVA) vector set to provide the homology repair template, this system is flexible and can readily be ported to a multitude of Gram-negative hosts. Besides genome editing, the Cas3 system can also be used as an effective and universal tool for vector curing. This is achieved by introducing a spacer that targets the origin-of-transfer, present on the majority of established (SEVA) vectors. Based on this, the Cas3 system efficiently removes up to three vectors in only a few days. As such, this curing approach may also benefit other genomic engineering methods or remove naturally occurring plasmids from bacteria.

**KEYWORDS** *Pseudomonas*, *Pseudomonas aeruginosa*, *Pseudomonas putida*, CRISPR-Cas, genome editing, vector curing, SEVA, Cas3

The *Pseudomonas* genus comprises a variety of aerobic, Gram-negative bacteria that are ubiquitous in nature, playing diverse biological roles that range from plant growth promotion to bioremediation and pathogenicity (1). The genus contains over 140 species (www.catalogueoflife.org) and is one of the most ecologically and medically

Address correspondence to Rob Lavigne, rob.lavigne@kuleuven.be, or Hanne Hendrix, hanne.hendrix@kuleuven.be.

The authors declare no conflict of interest.

See the funding table on p. 14.

 10.1128/spectrum.02707-23 **1**

important groups of bacteria. This includes the well-known antibiotic resistant pathogen *Pseudomonas aeruginosa*, which is responsible for infections in immunocompromised patients (2), and the biochemical versatile species *Pseudomonas putida* involved in industrial processing (3). Therefore, robust engineering tools for *Pseudomonas* cannot only support fundamental discoveries but also modify pathogenicity, improve production yield, and enable development of microbial cell factories (4, 5).

Diverse engineering systems are available to modify the genomes of *Pseudomonas* species. While the transposon-based systems insert DNA sequences in a random (e.g., Tn5 transposon) or site-specific manner (e.g., Tn7 transposon) and cannot delete genes (only disrupt them) (6, 7), the homologous recombination methods with integrative plasmids, such as the two-step allelic exchange and I-SceI-mediated recombination, require two rounds of selection using chromosomal markers with often low recombination frequency to achieve scarless genome editing (8, 9). More efficient recombineering methods using heterologous recombinases which catalyze recombination between similar sequences (e.g., λ Red and RecET recombinase systems) or between specific recognition sites (e.g., Cre/lox and Flp/FRT systems) also involve an additional step for integrated selection marker removal, extending the engineering time and often leaving a scar behind in the genome (10–13).

In recent years, the CRISPR-Cas (clustered regularly interspaced short palindromic repeats and CRISPR-associated proteins) systems have been proven to efficiently engineer genomes in virtually all species (14–19). The systems comprise an RNA guide called CRISPR RNA (crRNA) with sequence complementary to the target DNA (spacer), guiding a nuclease to make a site-specific break. Since many bacteria lack non-homologous end joining (NHEJ) to repair this break, a DNA repair template has to be provided to restore the defect by homologous recombination (20). Alternatively, the CRISPR-Cas system can be used as a counter-selection tool after recombineering or homologous recombination (21). Similar to other organisms, the most well-known engineering systems for *Pseudomonas* are based on single-subunit Class II CRISPR systems. These include the Type II CRISPR-Cas9 system of *Streptococcus pyogenes* (*Sp*Cas9) (21–27) and the Type V CRISPR-Cas12a from *Francisella novicida* (*Fn*Cas12a) (23, 28). However, one notable exception is *P. putida*, in which these CRISPR-Cas9-based systems with homology-directed repair (HDR) show reduced efficiency for genomic integration. This observation is probably due to the endogenous NHEJ of *P. putida*, which facilitates evasion of the targeted modification (29). Consequently, CRISPR-Cas9-based systems are solely used for deletion or as counter-selection tools in this host.

The Class I Type I system CRISPR-Cas3, on the other hand, consists of a multi-subunit complex and has the advantage to be the most prevalent in nature, enabling engineering with endogenous systems, and combines nuclease and helicase activities, degrading DNA processively and thus allowing larger deletions (30, 31). Recently, Csörgő et al. (31) exploited the Type I-C CRISPR-Cas3 system from *P. aeruginosa* (*Pae*Cas3c) for heterologous genome engineering in various microbial species, obtaining genome-scale deletions with random and programmed size and recombination efficiencies surpassing those of the *Sp*Cas9-based system. Also, in combination with heterologous recombinases, the *Pae*Cas3c CRISPR-Cas3 system has been successfully applied for genome engineering in multiple *Pseudomonas* species (32). Moreover, CRISPR-Cas3 has been introduced as the base editing tool CoMuTER, for targeted *in vivo* mutagenesis in yeast (33).

One major hurdle of CRISPR-Cas-assisted methods as well as other commonly used engineering techniques is the use of auxiliary plasmids, which need to be removed from the bacterial cells after engineering. Well-known curing systems rely on counter-selectable markers, repeated passaging of the cells, the use of tractable vectors, DNA intercalating agents, or conditional origins-of-replication (34–38). Nevertheless, these methods are often time-consuming, laborious, and not effective in some bacteria; can introduce off-target genomic mutations; or require specific vectors and conditions for their functionality (38–42). To avoid these issues, CRISPR-Cas-based plasmid curing

systems showed to be promising. Indeed, a recently developed CRISPR-Cas9-assisted curing system (pFREE) showed efficiencies between 40% and 100% for the major classes of vectors used in molecular biology, including Standard European Vector Architecture (SEVA) vectors, by targeting conserved sequences within origins-of-replication in multiple bacterial backgrounds (43, 44).

In this study, an efficient scarless genome editing and plasmid curing method based on CRISPR-Cas3 was developed for *Pseudomonas*. The system consists of the all-in-one pCas3cRh targeting plasmid designed by Csörgő et al. (31) combined with the SEVA vectors for homologous directed repair and curing, resulting in a straightforward, efficient, and universal system for genomic deletion and integration. The applicability of the method is demonstrated in *P. putida* KT2440 and SEM11 (45) and *P. aeruginosa* PAO1. Moreover, the system has been expanded by making it Golden Gate compatible, adding several antibiotic markers, and including a fluorescent marker to facilitate the screening procedure.

## RESULTS AND DISCUSSION

### An overview of the CRISPR-Cas3-based engineering approach for *Pseudomonas*

A CRISPR-Cas3-based engineering method was developed, which enables the creation of genomic deletions, insertions, or substitutions in the *Pseudomonas* genome in an efficient and flexible manner. In general, the Cas genes (*cas3*, *cas5*, *cas7*, and *cas8*) and crRNA with spacer sequence are all located on the pCas3cRh vector under the control of the RhaRS/$P_{RhaBAD}$ inducible system. Guided by the crRNA, the Cas3 enzyme creates a targeted cut in the genomic DNA upon induction with rhamnose. After cleavage, the damaged genome will be restored by HDR to create the desired genomic modification. To perform the HDR, a homology repair template is provided on vector pSEVA231 (Km$^R$, for *P. putida*) or pSEVA131 (Cb$^R$, for *P. aeruginosa*). The design of the repair template determines the prospective modification of the genome, namely, a deletion, insertion, or substitution. It is important to note that any canonical SEVA vector can serve as a carrier for the repair template, which allows the user to select a backbone with their preferred antibiotic marker and origin of replication for the application in mind and allows compatibility with any Gram-negative host (46). A notable exception are vectors carrying the *pRO1600/ColE1* origin of replication, as this is the replicative origin of the pCas3cRh vector (Table S2). Finally, after verification of the correct genomic modification with PCR and sequencing, the pCas3cRh and pSEVAX3-HDR vectors are cured from the host by introduction of pSEVA52-oriT. This vector expresses a spacer sequence targeting the *oriT* (origin-of-transfer) site, which is located on all SEVA plasmids (including itself) as well as many other established vectors and will enable the swift restriction and removal of the helper vectors (Fig. 1).

### The CRISPR-Cas3-based engineering system enables efficient genomic engineering of *P. putida*

In the following section, the engineering method will be described and illustrated in detail by means of an integration example in *P. putida* KT2440 and *P. putida* SEM11. More specifically, an expression construct consisting of $P_{14c}$-BCD22-phi15lys(G3RQ) is integrated in locus PP_5388 in both hosts, resulting in low, constitutive production of phi15 lysozyme (G3RQ) (Fig. 2) (47).

First, a PAM site is selected in proximity of the target, which will be the recognition site of the Cas3 enzyme. In general, the recognition site of the Cas3 system is defined as a 5′-AAG-3′ PAM with an upstream protospacer on the sense strand. However, in this work, we use the inverted terminology consistent with the work of Csörgő et al. (31): a 5′-TTC-3′ PAM in combination with a downstream protospacer located on the antisense strand. The PAM sequence is preferably located within the sequence that is to be deleted or substituted, or, in case of an integration, within 15 bp of the integration site. If no suitable

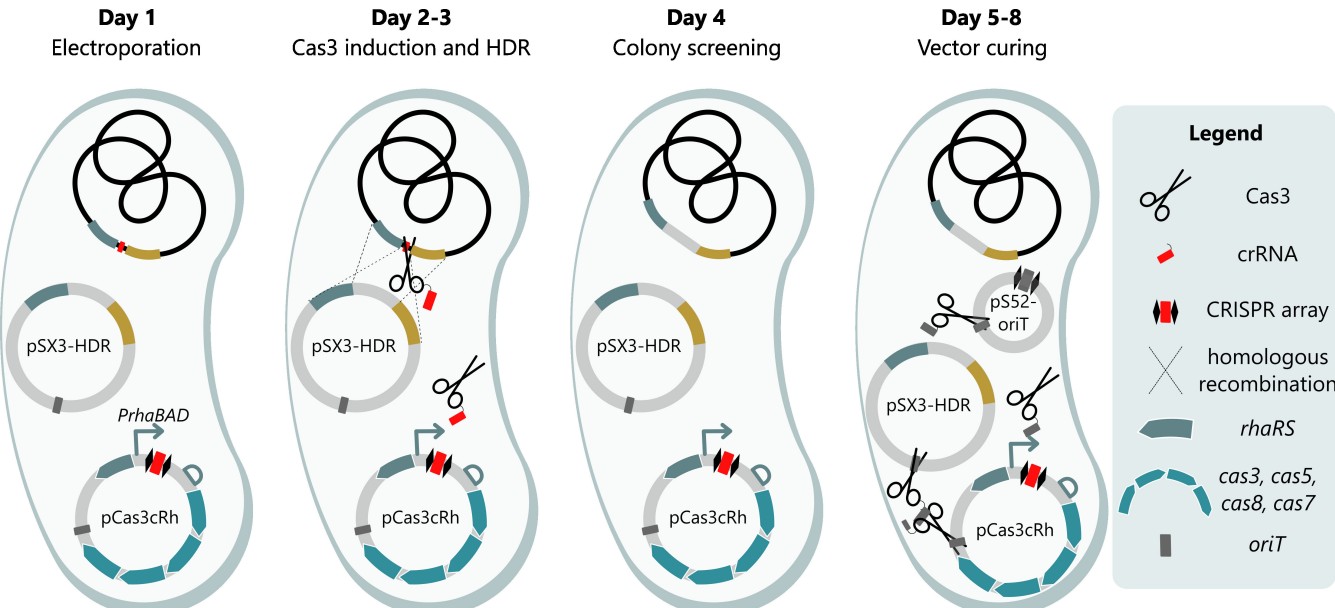

**FIG 1** General overview of the CRISPR-Cas3-based engineering method in *Pseudomonas*, as illustrated for *P. putida*. Day 1: the pCas3cRh vector with spacer sequence (red) for the target site and pSEVA231 with the repair template for homology-directed repair (pSX3-HDR) are introduced in Pseudomonas by means of electroporation. Days 2–3: on day 2, multiple colonies are inoculated together in growth medium supplemented with rhamnose to induce expression of the Cas3 system. The Cas3 enzyme cleaves the genomic DNA at the target location, i.e., the protospacer (marked in red), after which the dsDNA break will be repaired by homologous recombination using the repair template. After overnight induction, a dilution streak on lysogeny broth (LB) agar is performed on day 3. Day 4: multiple single colonies of the dilution streak are analyzed by PCR and Sanger sequencing, to verify the presence of the desired genomic modification. Correct mutants are grown overnight to start the vector curing process. Days 5–8: on day 5, the overnight cultures are transformed with pSEVA52-oriT (pS52-oriT), which carries a CRISPR array with a spacer sequence targeting the origin-of-transfer broadly used on plasmids. Similar to that in days 2–3, expression of the Cas3 system is induced on day 6, which will cleave all vectors and lead to efficient curing of the host strain. After a dilution streak on day 7 and overnight incubation, correct vector curing is verified on day 8 by streaking individual colonies on all antibiotics separately that were used to select the vectors. A similar method is applied for *P. aeruginosa*. However, no rhamnose is required to induce the system, and pSEVA131 is used for homology-directed repair.

PAM site is available in these regions, a site within the neighboring sequences of the genomic modification can be used as well, but the PAM site (or protospacer sequence) should be removed from the homology arms in later steps. The selected PAM site determines the spacer sequence, which is located directly downstream of the TTC trinucleotide, has a length of 34 bp, and should not have significant homology to secondary sequences in the genome as predicted by Blastn. The selected spacer sequence can be efficiently integrated in pCas3cRh by Golden Gate cloning with Type IIs restriction enzyme BsaI, as explained in Materials and Methods. For the example of integration in PP_5388 in *P. putida*, a PAM site was selected 1 bp upstream of the intended integration site (Fig. 2b) and the downstream spacer 5′-AGATCATGGTAACCCCG GCCGCTGGAGCCATTTC-3′ was successfully cloned into pCas3cRh to yield pCas3cRh-PP_5388 (Tables S1 and S2; Fig. S7).

After construction of the pCas3cRh-spacer vector, a second vector with the repair template needs to be assembled. Any canonical SEVA vector can be used for this purpose; however, for this work, we selected pSEVA231 (*P. putida*) and pSEVA131 (*P. aeruginosa*) due to their medium-copy number origin and appropriate resistance marker for the respective isolates. For deletions, the repair template simply consists of two joined homology arms, identical to the sequences directly up- and downstream of the region to be deleted. For integration or substitution, on the other hand, the repair template consists of the desired insertion or substitution, flanked by the up- and downstream homology arms. Homology arms of around 500 bp are preferred. However, shorter arms of ±280 bp have shown to work as well. It is important to note that selected recognition sites within the homology arms should be removed from the repair

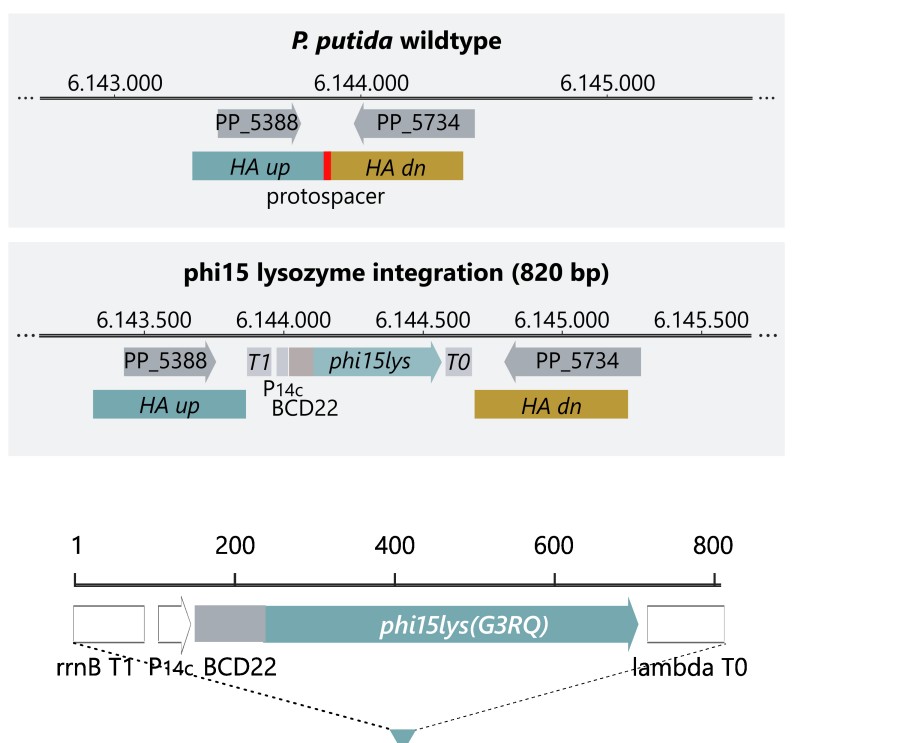

**FIG 2** Genomic integration of expression cassette P$_{14c}$-BCD22-phi15lys(G3RQ) in locus PP_5388 in *P. putida*. (a) Integration cassette P$_{14c}$-BCD22-phi15lys(G3RQ) has a total length of 820 bp. This phi15 lysozyme mutant G3RQ was optimized to inhibit the activity of the T7-like RNA polymerase (RNAP) of phage phi15, to reduce basal expression of this RNAP in uninduced conditions, similarly to the established pET system (47). The PP_5388 was previously identified as a locus that results in low expression levels of integrated sequences (48). As such, low levels of the phi15 lysozyme (G3RQ) will inhibit basal concentrations of the phi15 RNAP, while leaving sufficient active RNAP molecules upon induction of the pET-like system. (b) The protospacer adjacent motive (PAM) site (indicated in bold) lies 1 bp upstream of the integration site (triangle) where the expression cassette will be integrated. The protospacer sequence (highlighted in red) is defined as the first 34 bp directly downstream of the PAM site. After correct integration, the PAM and protospacer sequence will be separated from each other. Without the adjacent PAM site, the protospacer is no longer a suitable cleavage site for the Cas3 enzyme complex.

template, either by deletion, PAM mutation, or protospacer/PAM interruption. For the selected integration in PP_5388, homology arms of 550 bp each were amplified from the genome *of P. putida* and ligated to flank the integration cassette in pSEVA23-PP_5388, using Golden Gate cloning with Type IIs restriction enzyme BsaI (Fig 2a; Fig. S7, Tables S1 and S2). In this particular example, the integration site is located within the protospacer region. As such, the protospacer/PAM sequence will be interrupted upon correct integration and will no longer be cleaved by the Cas3 enzyme complex (Fig. 2b). If the integration site was not located within the protospacer sequence, the PAM site should be mutated in the repair template to avoid cleavage of the HDR vector and the genome upon integration.

Following the vector construction, both the pCas3cRh-spacer and the template vector are simultaneously introduced into the *Pseudomonas* host by co-electroporation. If the efficiency of the co-electroporation is insufficient, the vectors can be introduced consecutively by first introducing the repair template followed by pCas3cRh-spacer, adding one additional day to the protocol. For the PP_5388 integration, both *P. putida* KT2440 and *P. putida* SEM11 were successfully co-transformed with 0.25*10$^3$ and 2.5*10$^3$ CFU/µg DNA, respectively, and no morphological differences of the colonies were observed in comparison to electroporation with empty control vectors. Furthermore, pCas3cRh-PP_5388 was also successfully introduced separately, indicating that little to

no basal expression occurs from the Cas3 system in *P. putida* and that the RhaRS/$P_{rhaBAD}$ expression system is tightly regulated. To confirm this, 24 co-transformants of *P. putida* KT2440 and *P. putida* SEM11 were analyzed by PCR with primers binding on the genome outside the homology arms, showing that none of the co-transformants had the desired insertion before induction of the CRISPR-Cas3 system (Fig. S1).

To induce the CRISPR-Cas3 system, several co-transformants were pooled and used to inoculate 20 mL LB medium with the required antibiotics and 0.1% (wt/vol) rhamnose. The cultures were then incubated overnight at the appropriate temperature. The following day, a dilution streak of the induced overnight culture was performed on agar plates with the appropriate antibiotics and grown until visible colony formation the following day. For the PP_5388 integration example, again, 24 colonies of *P. putida* KT2440 and *P. putida* SEM11 were subjected to PCR with primers binding outside the homologous arms on the genome. Interestingly, after induction with rhamnose, 83% of *P. putida* KT2440 colonies and 88% of *P. putida* SEM11 colonies showed an amplicon length correlating to correct integration of the *$P_{14c}$-BCD22-phi15lys(G3RQ)* cassette (Fig. 3a; Fig. S2). In comparison for uninduced control samples, no integration was observed in any of the screened *P. putida* KT2440 or *P. putida* SEM11 colonies (Fig. 3a; Fig. S2). As such, the Cas3 system is able to efficiently perform genomic integrations in *P. putida* without the assistance of any recombineering approaches as required for integration with the Cas9 system (23, 49).

To put these engineering efficiencies into perspective, the same integration in *P. putida* was created using traditional homologous recombination (HR). More specifically, the two-vector system as described by Volke et al. (37) was employed, where the first vector, carrying the homology arms and desired modification, fully integrates in the genome in a first HR event. This event can be tracked by a green fluorescent reporter and antibiotic resistance marker on the integration vector. Next, a second vector supplies the I-SceI restriction enzyme, which will recognize and cut a unique restriction site within the integrated vector and force the second HR event, resulting in the desired genomic modification with loss of the fluorescent reporter and antibiotic marker. In this work, we successfully constructed integration vector pSNW2-PP_5388-*$P_{14c}$-BCD22-phi15lys(G3RQ)*, which integrated in the *P. putida* KT2440 and *P. putida* SEM11 hosts after electroporation. After overnight incubation of several transformants, the pSEVA62313S helper vector with constitutive expression of I-SceI was introduced into the hosts by electroporation. As recommended in the original protocol (37), the resulting colonies were transferred to a fresh LB agar plate by streaking to avoid mixed-phenotype colonies. The resulting colonies were screened for a successful second HR event by verifying the lack of green fluorescence, followed by a PCR with the same genomic primers as for the CRISPR-Cas3-based method. For *P. putida* KT2440, only one of 24 PCR-screened colonies contained a correct integrant, while for *P. putida* SEM11, no correct integrants were obtained (0/24) but it still appeared to have a mixed phenotype (Fig. 3a; Fig. S3). Therefore, the *P. putida* SEM11 strain carrying pSNW2-PP_5388-*$P_{14c}$-BCD22-phi15lys(G3RQ)* and pSEVA62313S helper vector was streaked twice more to allow additional time for the second HR event to occur. After a second PCR screen, 21% (5/24) of the screened colonies showed an amplicon length correlating to correct integration of the *$P_{14c}$-BCD22-phi15lys(G3RQ)* cassette (Fig. 3a; Fig. S3). Overall, the engineering efficiencies obtained by homologous recombination were much lower compared with the CRISPR-Cas3-based method and required significantly more handling time, due to consecutive electroporation of the vectors and multiple streaking steps.

## The CRISPR-Cas3 system cures itself with high efficiency in *P. putida* using an *oriT*-targeting spacer

After successful engineering of the host genome, cells need to be cured from the pCas3cRh and repair template vector for downstream processing. A universal CRISPR-Cas3-based curing concept was introduced, similar to the proven CRISPR-Cas9-based curing method for *Escherichia coli* and *P. putida*, which makes use of spacers targeting

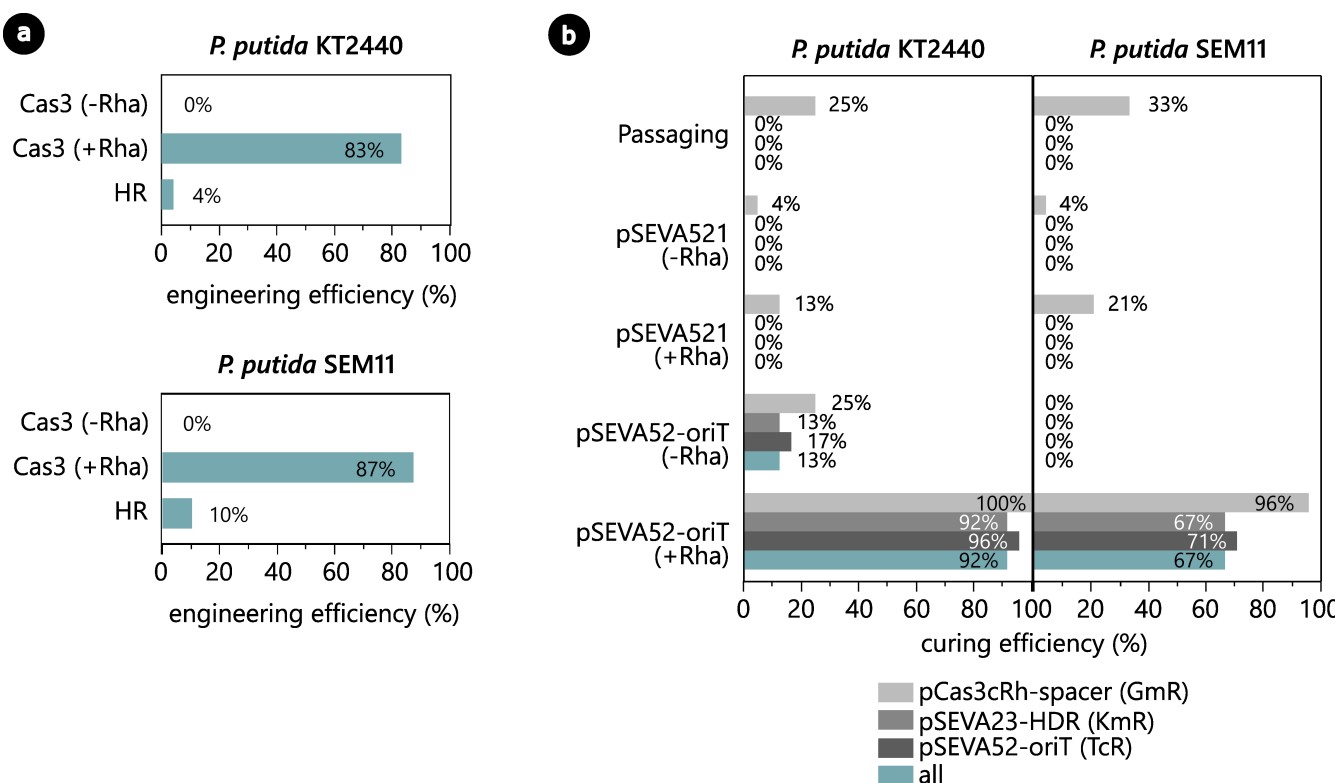

**FIG 3** (a) Engineering efficiencies of the integration of expression cassette $P_{14c}$-BCD22-phi15lys(G3RQ) in locus PP_5388 of *P. putida* KT2440 and *P. putida* SEM11 using the CRISPR-Cas3-based method (Cas3), with or without induction with rhamnose (±Rha), or via traditional homologous recombination (HR). (b) After engineering of *P. putida* with pCas3cRh-PP_5388 and pSEVA23-PP_5388, the strains are cured from the engineering vector by serial passaging or by CRISPR-Cas3-based curing using pSEVA52-oriT, with or without rhamnose induction (±Rha). As a negative control for the CRISPR-Cas3-based curing, an empty pSEVA521 vector was used instead of pSEVA52-oriT.

conserved regions of plasmid, i.e., the origins-of-replication (*oriR*) (43). As the cells in this work already contained the Cas3 system, it can simply be used to target itself by introducing crRNA with a self-targeting spacer. To this end, a universal spacer was designed, binding specifically to the *oriT* located on all SEVA plasmids and many other commonly used vectors for genome engineering, including pCas3cRh. This *oriT* spacer and crRNA were cloned into pSEVA521 under control of the $P_{RhaBAD}$ promoter (Tables S1 and S2; Fig. S7) and called pSEVA52-oriT.

The pSEVA52-oriT vector was introduced in the engineered *P. putida* KT-phi15lys and *P. putida* S-phi15lys strains through electroporation, after which cells were plated on LB agar supplemented with gentamicin (pCas3cRh-PP_5388) and tetracycline (pSEVA52-oriT). In parallel, the same strains were electroporated with pSEVA521 as a negative control. The following day, several colonies of each strain were grown in $LB^{Gm10/Tc10}$ medium with 0.1% (wt/vol) rhamnose to induce expression of the Cas3 system. After a dilution streak on LB medium without any antibiotics and overnight incubation, 24 colonies of each condition were screened on gentamicin (pCas3cRh-PP_5388), kanamycin (pSEVA23-PP_5388), and tetracycline (pSEVA52-oriT) to assess the curing efficiency. In the presence of the *oriT* spacer, 91.6% and 66.7% of colonies were fully cured of all vectors for *P. putida* KT2440 and *P. putida* SEM11, respectively (Fig. 3b). This is in sharp contrast to the control samples with the empty pSEVA521 vector, of which all of the screened colonies still contained at least two of the three vectors. Furthermore, the engineered strains were also subjected to serial passaging for the same amount of time as required for the CRISPR-Cas3-based curing. Four passages were performed over 3 days, after which none of the screened colonies were cured from the pCas3cRh and pSEVA23-PP_5388 vectors (Fig. 3b). These results show that the CRISPR-Cas3-system

is able to efficiently target itself and other vectors in the same cell, with enhanced efficiencies compared with the original CRISPR-Cas9-based curing approach (53% curing efficiency in *P. putida*) (43).

After successful vector curing, two biological replicates of *P. putida* KT-phi15lys and *P. putida* S-phi15lys were subjected to whole-genome sequencing. No deletions or insertions were detected, except for four point mutations in the *P. putida* KT-phi15lys replicates (Tables S5 and S6) and two-point mutations in both *P. putida* S-phi15lys replicates (Tables S7 and S8). All observed mutations were located outside of the integration cassette.

## Application examples: efficient genomic deletion of three different targets in *P. aeruginosa*

To show that the CRISPR-Cas3-based engineering method is also functional in other hosts, three separate genomic deletions were created in the genome of *P. aeruginosa* PAO1. More specifically, three sets of spacers and repair templates were designed to delete the entire coding sequences of *fleS*, PA_2560, and *prpL* (Fig. 4a). After successful construction of all six vectors, the corresponding pCas3cRh-spacer and pSEVA13-HDR vectors were simultaneously introduced in *P. aeruginosa* PAO1. Surprisingly, visible colonies only appeared after a 2-day incubation period for PA_2560 and *prpL*, while a control electroporation with the empty pCas3cRh or pSEVA131 vector resulted in colony formation overnight. For the *fleS* deletion, even after multiple days of incubation, no colonies grew on plates of the co-electroporation and the pSEVA13-HDR and pCas3cRh vectors had to be introduced consecutively. This is in sharp contrast to the results with *P. putida*, where co-electroporation resulted in normal colony formation after a single night of incubation.

This indicates that *P. aeruginosa* shows retarded cell growth after introducing the engineering vectors, which points towards significant basal Cas3 expression from the RhaRS/$P_{rhaBAD}$ system in this host. To confirm this hypothesis, the CRISPR-Cas3 system was not induced with rhamnose, but the transformants were analyzed directly after electroporation with PCR for the genomic deletion. Indeed, for all three deletions, at least 88% of the screened colonies already contained the desired deletion (Fig. 4b; Fig. S4 to 6). This confirms that in *P. aeruginosa*, the basal expression of the CRISPR-Cas3-system is sufficient for genome engineering and rhamnose induction is not required. Other inducible systems could be explored to create a more stringent regulation of the Cas3 system in *P. aeruginosa*.

After successful deletion of the targeted genes, all three deletion mutants were cured from the respective pCas3cRh-spacer and pSEVA13-HDR vectors using the *oriT*-targeting approach. The pSEVA52-oriT vector was introduced in all strains, after which the transformants were grown overnight in antibiotic-free medium without rhamnose. The following day, a dilution streak was performed and the resulting colonies were screened for sensitivity against gentamycin (pCas3cRh-spacer), carbenicillin (pSEVA13-HDR), and tetracycline (pSEVA52-oriT). Both the pCas3cRh-spacer and pSEVA52-oriT vectors were cured very effectively, with a curing efficiency ranging from 64% to 100% (Fig. 4c). The pSEVA13-HDR vectors, on the other hand, were still present in the majority of the screened colonies, resulting in a rather low curing efficiency rates of 17%, 42%, and 50% for the Δ*fleS*, Δ*prpL*, and ΔPA_2560 mutants, respectively. This difference in curing efficiency between the vectors could be explained by the fact that the pCas3cRh vector exerts a negative selection pressure upon itself once pSEVA52-oriT is present, in contrast to the other two vectors. Furthermore, the pSEVA52-oriT vector contains the low-copy RK2 *oriR*, while the pSEVA13-HDR vector carries the medium-copy BBR1 *oriR*, which could explain why the pSEVA52-oriT origin is more efficiently cured than its pSEVA13-HDR counterpart. To further improve the flexibility of the system and the efficiencies achieved, additional spacers could be included on pSEVA52-oriT to target a variety of *oriRs*, shown to be effective in previous work (43).

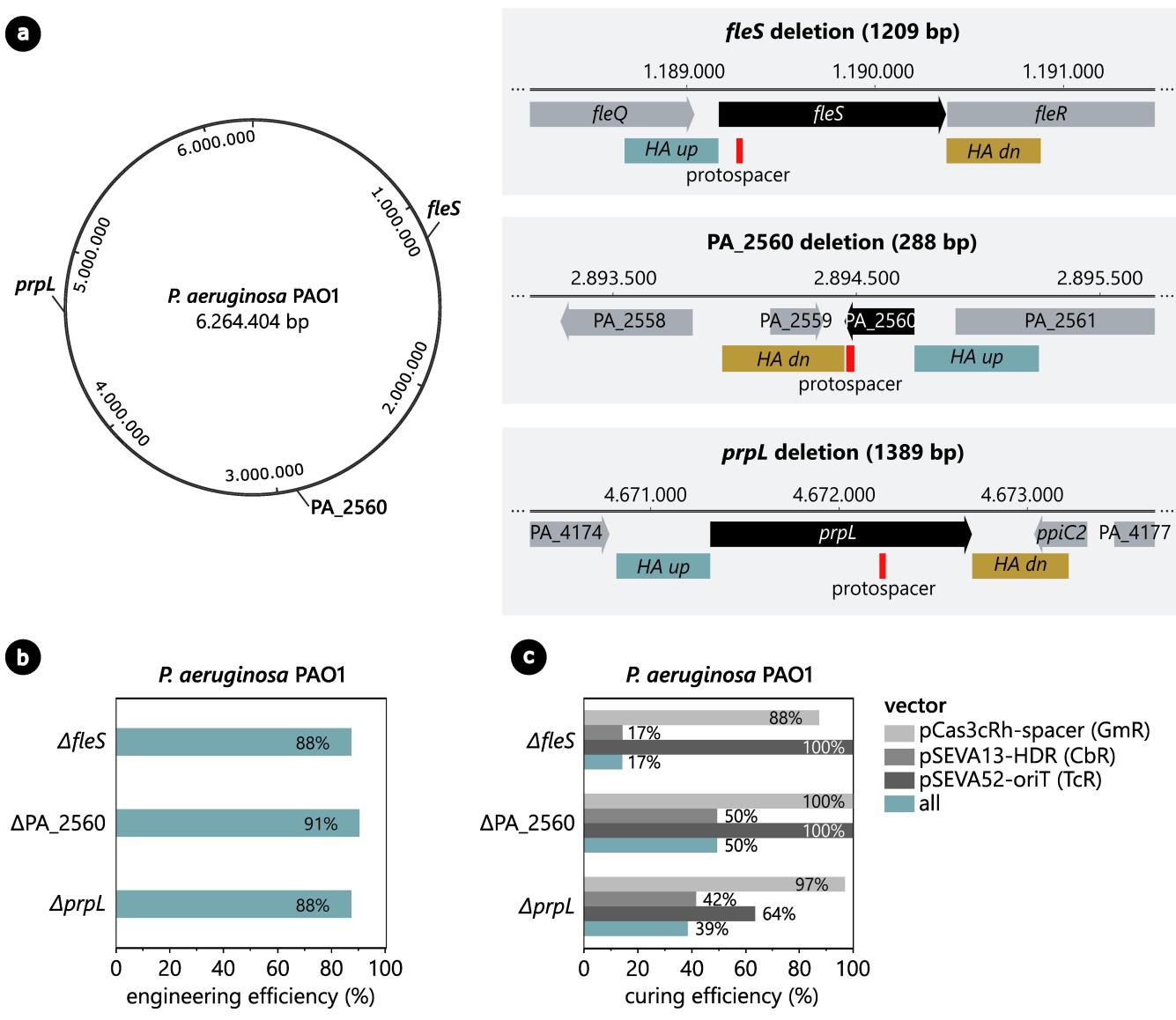

**FIG 4** (a) Three separate genomic deletions are created in the *P. aeruginosa* PAO1 genome, namely, the entire coding regions of fleS, PA_2560, and prpL (indicated in black). The position of the protospacer in the genes is marked in red, and the upstream and downstream homology arm are indicated in cyan and ochre, respectively. (b) Engineering efficiency of the CRISPR-Cas3-based engineering method to create the fleS, PA_2560, and prpL deletions. (c) Curing efficiency of the CRISPR-Cas3-based curing method for pCas3Rh-spacer, pSEVA13-HDR, and pSEVA52-oriT in the *P. aeruginosa* PAO1 ΔfleS, ΔPA_2560, and ΔprpL deletion mutants.

## An easy-to-clone vector set with a broad range of antibiotic markers further improves the CRISPR-Cas3-based engineering method

Two vector sets were created to facilitate cloning of the homology arms and to allow compatibility of the CRISPR-Cas3 engineering system with different hosts or experimental set-ups requiring different antibiotic selection markers. The first vector set for HR cloning comprises five pSEVAX3-GG vectors, all encoding a Golden Gate cassette and different antibiotic markers (Fig. 5a). The Golden Gate cassette consists of an msfGFP (monomeric superfolder green fluorescent protein) reporter driven by a strong constitutive promoter ($P_{14g}$) (50) and flanking BsaI recognition sites (Fig. 5c). The second vector set, on the other hand, is derived from pCas3Rh and holds five pCas3-Ab vectors with different antibiotic markers (Fig. 5b). As such, the user has the possibility to select their favorite vector combination for the genomic engineering experiment in mind.

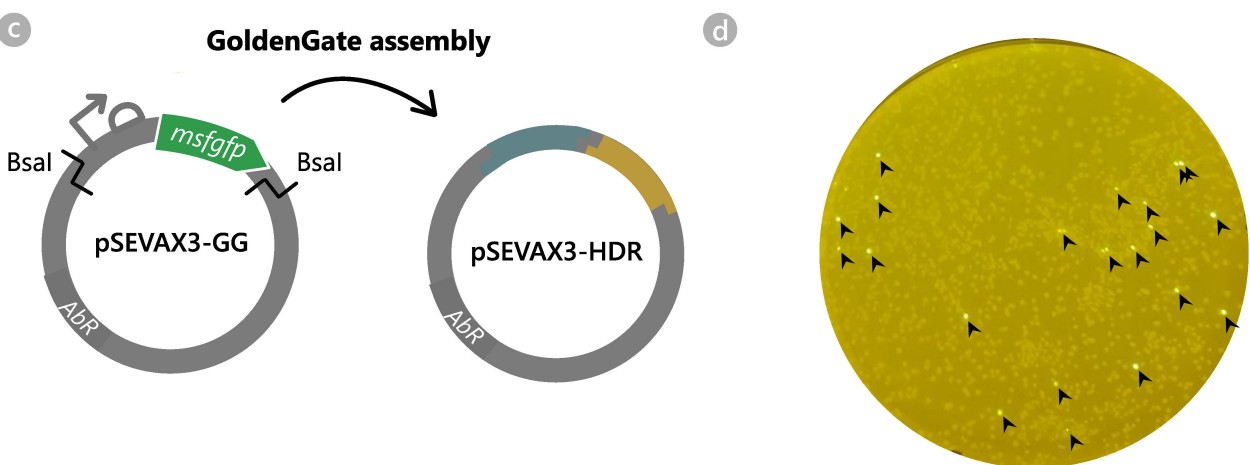

**FIG 5** (a) Vector set of pSEVAX3-GG for Golden Gate cloning of the homology arms for CRISPR-Cas3 engineering. All vectors are identical, except for the antibiotic selection marker. The vectors are equipped with a Golden Gate cassette, consisting of an msfGFP reporter flanked with BsaI recognition sites. (b) Vector set of pCas3cRh-derived vectors for CRISPR-Cas3-based genome engineering. All vectors are identical, except for the antibiotic selection marker. (c) Golden Gate assembly of the homology arms into pSEVAX3-GG vectors. The Golden Gate cassette with the msfGFP reporter is substituted for the homology arms for HDR (teal and yellow). (d) LB agar plate on an ultrabright-LED transilluminator (470 nm): *E. coli* after transformation with assembled pSEVAX3-HDR vector (Golden Gate reaction mix). Colonies which are false-positive and thus contain the original pSEVAX3-GG vector are easily identified by their msfGFP fluorescence (indicated with a black arrow).

## Conclusions and perspectives

A novel CRISPR-Cas3-assisted editing method is presented for *Pseudomonas*, showcasing high efficiency for genomic integration or deletion in *P. putida* and *P. aeruginosa* (>83%). The use of a second, universal SEVA vector for homology-directed repair along with the relatively large targeting plasmid (11.5 kbp), enables to simplify the plasmid construction and increase the efficiency compared to the one-vector system of Csörgo et al. (31). Moreover, a vector set of both plasmids with multiple antibiotic markers allows for application in various Gram-negative hosts and different designs. In addition, due to the inherent ability of the Cas3 enzyme to cleave and thereby cure plasmids from the host strain, a third plasmid containing a spacer that targets the *oriT* is introduced to rapidly and effectively remove all helper vectors in only a few days, with up to 100% curing efficiency. Although the curing concept is similar to previously described CRISPR-Cas9 system (43), enhanced efficiencies using the CRISPR-Cas3 system are obtained. As such, the described approach is an elegant addition to the CRISPR-Cas-based engineering toolbox for *Pseudomonas*. Besides exemplified cases, the protocol has been successfully applied for multiple more cases, ranging from genomic insertions, substitutions and deletions, including the deletion of a 10 kbp prophage region from the *P. aeruginosa* PAO1 genome.

Apart from their use in genomic engineering, the pCas3-AbR and pSEVA52-oriT vectors can be used as a stand-alone tool for vector curing of any synthetic or

naturally-occurring plasmid in *Pseudomonas*. This can be achieved by introducing multiple spacers targeting conserved sequences of plasmids, as described by Lauritsen et al. (43). By integrating the *oriT* spacer on the pCas3-AbR vector under control of a strictly regulated promoter, only a single vector would be required for curing purposes.

In future work, the possibilities of the Cas3 editing approach can be further explored, e.g., by creating larger genomic alterations or by performing several genomic edits simultaneously, by providing more than one spacer and repair template on the pCas3-AbR and pSEVAX3-GG plasmids. Additionally, due to the flexibility of the proposed vectors sets, namely the pSEVAX3-GG and pCas3-AbR sets, the functionality of the Cas3 approach can readily be investigated in related *Pseudomonas* species or other Gram-negative strains.

## MATERIALS AND METHODS

### Strains and media

All strains used in this work are listed in Table S3. Overall, vector construction was performed in *E. coli* TOP10 and CRISPR-Cas3-based engineering was carried out in *P. putida* KT2440, *P. putida* SEM11 and *P. aeruginosa* PAO1 (Table S3). All strains were cultured in standard LB medium or agar, supplemented with the appropriate antibiotics: Gm10 (*E. coli* and *P. putida*) or Gm50 (*P. aeruginosa*), Km50 (*E. coli* and *P. putida*), Ap100 (*E. coli*), Cb200 (*P. aeruginosa*), Tc10 (*E. coli* and *P. putida*), or Tc60 (*P. aeruginosa*). *P. putida* was incubated at 30°C, whereas *E. coli* and *P. aeruginosa* were incubated at 37°C.

### Vector construction: pCas3cRh-spacer

A spacer sequence was identified in the target region and introduced in the pCas3cRh vector by Golden Gate ligation. First, the spacer was created by annealing two primers: (i) GAAAC-[spacer sequence]-G and (ii) GCGAC-[reverse complement of spacer sequence]-G. The primers used in this work are listed in Table S1. The annealed primer pair (50 ng) was combined with pCas3cRh (100 ng), T4 DNA ligase (1 U, Thermo Scientific), BsaI (10 U, Thermo Scientific), and 1× DNA ligation buffer (Thermo Scientific), after which the reaction mixture was subjected to 30 restriction-ligation cycles (37°C for 2 min; 16°C for 3 min). Next, the reaction mixture was introduced in *E. coli* TOP10 via heat-shock transformation (51). After overnight incubation on LB^Gm10 agar, multiple transformants were screened for the presence of the spacer using DreamTaq Green PCR (Thermo Scientific) with primers pCas3cRh_F/R (Table S1). Amplicons with the expected length were Sanger sequenced (Eurofins Genomics, Germany), and corresponding vectors were purified with the GeneJet Miniprep Kit (Thermo Scientific) (Table S2).

### Vector construction: pSEVAX3-HDR

The template for HDR is provided on pSEVA131 (*P. aeruginosa*) or pSEVA231 (*P. putida*), further referred as pSEVAX31, and assembled by Golden Gate cloning. First, the upstream and downstream homology arms (HA up and dn), desired insert (for integrations only), and vector backbone were amplified with Phusion polymerase (Thermo Scientific) with tailed primers, to introduce the BsaI recognition site and BsaI restriction site for Golden Gate ligation (Table S1). Of note, instead of creating the homology arms by PCR, a gene block containing the HDR template flanked by BsaI restriction sites can be synthesized synthetically. The use of synthetic gene blocks might be advantageous for difficult-to-clone fragments or for fragments that require (multiple) sequence alterations, e.g., PAM or protospacer sequence alteration in case of substitution or insertion or codon optimization. All nucleotide sequences of used HAs and inserts in this work are provided in Table S4. The BsaI restriction sites are designed to allow specific annealing of HA up – (insert) – HA dn in the pSEVAX3 amplicon. The amplicons of the homology arms (50 ng) each and insert (50 ng) were combined with the PCR-amplified pSEVAX3 backbone (100 ng), T4 DNA ligase (1 U, Thermo Scientific), BsaI (10 U, Thermo Scientific), and 1×

DNA ligation buffer (Thermo Scientific), after which the reaction mixture was subjected to 50 restriction-ligation cycles (37°C for 2 min; 16°C for 3 min). Next, the reaction mixture was introduced in *E. coli* TOP10 via heat-shock transformation (51). After overnight incubation on LB^Km50 or LB^Ap100 agar, multiple transformants were screened for the presence of the template using DreamTaq Green PCR (Thermo Scientific) with primers SEVA_PS1/2 (Table S1). Amplicons of the expected length were Sanger sequenced (Eurofins Genomics, Germany), and the corresponding vectors were purified with the GeneJet Miniprep Kit (Thermo Scientific) (Table S2).

## Vector construction: pSEVA52-oriT

Vector pSEVA52-oriT was constructed in two steps. First, a pCas3cRh vector with *oriT* spacer was constructed as described above, with oriT_spacer_F/R (Tables S1 and S2). Second, the $P_{RhaBAD}$ promoter and CRISPR array with *oriT* spacer were amplified from pCas3cRh-oriT with tailed primers oriT_Cas3_F/R and the pSEVA521 backbone was linearized with Phusion PCR (Thermo Scientific) with tailed primers oriTcas3_SEVA_F/R (Table S1). Both amplicons were annealed by Golden Gate ligation, as described above for the construction of pSEVAX3-HDR vectors. Multiple *E. coli* TOP10 transformants were screened for the presence of the *oriT* CRISPR array using DreamTaq Green PCR (Thermo Scientific) with primers SEVA_PS1/2 (Table S1). Amplicons of the expected length were Sanger sequenced (Eurofins Genomics, Germany), and the final pSEVA52-oriT vector was purified with the GeneJet Miniprep Kit (Thermo Scientific) (Table S2).

## Vector construction: pCas3-XX and pSEVAX3-GG vector sets

To create Cas3 bearing plasmids with different antibiotic selection markers (pCas3-Amp, pCas3-Km, pCas3-Sm, pCas3-Gm, and pCas3-Apr; Table S2), pCas3cRh was amplified with primer pair pCas3_Ab_F/R (Table S1) and the antibiotic selection cassettes were amplified from canonical SEVA plasmids (46) with the primer pair Ab_F/R. The antibiotic selection fragments were ligated with the pCas3cRh amplicon by USER cloning (52). Following transformation of *E. coli*, colony PCR, and plasmid purification as described above, correctness of plasmids was confirmed by whole plasmid sequencing (Plasmid-saurus, Oregon, USA).

For the creation of the pSEVAX3-GG vector set, pSEVA131 was amplified with the primer pair pSX31_GG_F/R, while a fragment carrying *msfgfp* under the constitutive promoter 14g with BCD2 was amplified from pBG42 (50) with primer pair P14g-BCD2-GFP_F/R. Fragments were merged by USER cloning into pSEVA13-GG, and the correctness of the plasmid inserts was confirmed by Sanger sequencing with SEVA_PS1/2. The overhangs created by BsaI were designed for optimal cloning efficiency (53). Subsequently, the antibiotic cassette of the plasmid was exchanged by USER cloning to create pSEVA23-GG, pSEVA43-GG, pSEVA63-GG, and pSEVA83-GG (Table S2). The vector, linearized with the primer pair pSX31_Ab_F/R, was merged with the same fragments used for antibiotic cassette exchange for pCas3cRh. Correct vector assembly was verified with nanopore, whole plasmid sequencing. Finally, vectors pCas3-ApR and pSEVA83-GG were subjected to full linearization with a tailed primer (ApR_BsaI_F/R) and religated with USER cloning, to remove an undesired BsaI recognition site from the *apR* gene.

## Electroporation

*P. putida* and *P. aeruginosa* were electroporated according to the protocol described by Choi et al. (54). In brief, per sample, a 3-mL overnight culture was washed three to five times with 2 mL sterile 10% (wt/wt) sucrose solution to create electrocompetent cells. After the washing steps, 20–50 ng plasmid DNA was added to a 100-µL cell aliquot and electroshocked at 200 ohm, 25 µF, and 1.8 kV or 2.0 kV for *P. aeruginosa* and *P. putida*, respectively. For co-electroporations, 100 ng of each plasmid was added to the cell aliquot together and electroshocked in the same manner. After cell recovery for 1.5 h in 900 µL LB or SOC medium at the appropriate temperature, different volumes

of the aliquot (10 µL, 50 µL, 250 µL, and the rest) were plated on selective LB agar and incubated overnight, unless specifically mentioned otherwise.

## CRISPR-Cas3-based engineering and vector curing in *P. putida*

Overnight cultures of *P. putida* were co-electroporated with pCas3cRh-PP_5388 and pSEVA23-PP_5388 as described above. After overnight incubation on LB$^{Km50/Gm10}$ agar, five colonies were inoculated together in 20 mL LB$^{Km50/Gm10}$ with 0.1% (wt/vol) rhamnose (Merck, CAS no. 10030–85-0) for induction of the CRISPR-Cas3 system and incubated overnight while shaking. The next day, a dilution streak of the 20 mL culture is performed on LB$^{Km50/Gm10}$ agar and again incubated overnight, after which 24 colonies were screened for correct genomic integration of the insert with DreamTaq Green PCR (Thermo Scientific) with primers PP5388_up/dn (Table S1). Amplicons of the expected length were Sanger sequenced (Eurofins Genomics, Germany), and the corresponding colonies were cured from pCas3cRh-PP_5388 and pSEVA23-PP_5388. For vector curing, overnight cultures were electroporated with pSEVA521-oriT and the CRISPR-Cas3 system is induced as mentioned previously, using overnight incubation with 0.1% (wt/vol) rhamnose followed by a dilution streak on LB medium without antibiotics. From the resulting plates, 24 colonies were streaked on LB, LB$^{Km50}$, LB$^{Gm10}$, and LB$^{Tc10}$ and incubated overnight to assess the successful vector curing by antibiotic sensitivity.

## CRISPR-Cas3-based engineering and vector curing in *P. aeruginosa*

Overnight cultures of *P. aeruginosa* were co-electroporated with pCas3cRh-spacer and pSEVA131-HDR as described above. For deletion of *fleS*, the co-electroporation did not result in colony formation, such that pSEVA13-FleS and pCas3cRh-FleS were introduced consecutively. After a 2-day incubation period on LB$^{Cb200/Gm10}$ agar, 14–24 colonies were screened for correct genomic deletion of the target gene with DreamTaq Green PCR (Thermo Scientific) with primers gene_up/dn (Table S1). Amplicons of the expected length were Sanger sequenced (Eurofins Genomics, Germany), and the corresponding colonies were cured from pCas3cRh-spacer and pSEVA13-HDR. For vector curing, overnight cultures were electroporated with pSEVA52-oriT and incubated overnight. The following day, 24 colonies were streaked on LB, LB$^{Cb200}$, LB$^{Gm50}$, and LB$^{Tc60}$ and incubated overnight to assess successful vector curing by antibiotic sensitivity.

## Whole-genome sequencing

The genomic DNA of the CRISPR-Cas3-engineered strains after vector curing was isolated using the DNeasy UltraClean Microbial Kit (Qiagen, Germany) according to the manufacturer's guidelines. The obtained DNA was sequenced with an Illumina platform (USA) and an Oxford Nanopore Technologies platform (UK) for long-read DNA sequencing. The Illumina DNA libraries were prepared using the Illumina DNA Prep Kit (USA) and the Nextera DNA CD Indexes (Illumina, USA). The average length of the DNA libraries was evaluated using Agilent Bioanalyzer 2100 and a High Sensitivity Kit (Agilent Technologies, USA), and the concentration of the DNA libraries was determined with a Qubit device (Thermo Fisher Scientific, USA). Next, the samples were pooled together for sequencing on the Illumina MiniSeq NGS platform. The MiniSeq Mid Output Kit (300 cycles) (Illumina, USA) was used for paired-end sequencing (2 × 150 bp), aiming for 800,000 reads per sample.

For Nanopore sequencing, the Rapid Barcoding Kit 24 V14 (Oxford Nanopore Technologies, UK) was used for library preparation. A maximum of 24 samples were pooled and sequenced on a R10.4.1 flowcell (Oxford Nanopore Technologies, UK). The raw Illumina and Nanopore reads were trimmed with Trimmomatic (55) or Porechop (56), respectively, after which they were assembled into complete circular genomes with Unicycler (57). Large deletions were visualized in IGV after Bowtie2 assembly (58), and single nucleotide polymorphism (SNP) analysis was performed with SNIPPY (59).

## ACKNOWLEDGMENTS

The pCas3cRh vector was kindly provided by prof. J. Bondy-Denomy (UCSF).

This project received funding from the European Research Council (ERC) under the European Union's Horizon 2020 Research and Innovation Programme (Grant Agreements 819800 and 814418), from the Fonds voor Wetenschappelijk Onderzoek Vlaanderen (FWO) as part of the CELL-PHACTORY Project (Grant G096519N), from the Novo Nordisk Foundation (Grant Agreements NNF10CC1016517 and NNF18CC0033664), and by a grant from KU Leuven (C1 project "ACES," C16/20/001).

## AUTHOR AFFILIATIONS

[1]Laboratory of Gene Technology, Department of Biosystems, KU Leuven, Leuven, Belgium
[2]The Novo Nordisk Foundation Center for Biosustainability, Technical University of Denmark, Lyngby, Denmark

## AUTHOR ORCIDs

Eveline-Marie Lammens ⓘ http://orcid.org/0000-0002-4946-4469
Kaat Schroven ⓘ https://orcid.org/0000-0002-3663-8343
Rob Lavigne ⓘ http://orcid.org/0000-0001-7377-1314
Hanne Hendrix ⓘ http://orcid.org/0000-0002-5432-9456

## FUNDING

| Funder | Grant(s) | Author(s) |
| --- | --- | --- |
| EC | Horizon 2020 Framework Programme (H2020) | 819800 | Kaat Schroven |
| | | Alison Kerremans |
| EC | Horizon 2020 Framework Programme (H2020) | 814418 | Daniel Christophe Volke |
| Fonds Wetenschappelijk Onderzoek (FWO) | G096519N | Eveline-Marie Lammens |
| Novo Nordisk Fonden (NNF) | NNF10CC1016517 | Daniel Christophe Volke |
| Novo Nordisk Fonden (NNF) | NNF18CC0033664 | Daniel Christophe Volke |
| KU Leuven (Katholieke Universiteit Leuven) | C16/20/001 | Hanne Hendrix |

## AUTHOR CONTRIBUTIONS

Eveline-Marie Lammens, Conceptualization, Data curation, Formal analysis, Investigation, Methodology, Validation, Visualization, Writing – original draft | Daniel Christophe Volke, Conceptualization, Data curation, Formal analysis, Investigation, Methodology, Writing – review and editing | Kaat Schroven, Data curation, Investigation, Writing – review and editing | Marleen Voet, Data curation, Investigation, Methodology | Alison Kerremans, Data curation, Investigation | Rob Lavigne, Conceptualization, Funding acquisition, Resources, Supervision, Writing – review and editing | Hanne Hendrix, Conceptualization, Data curation, Formal analysis, Investigation, Methodology, Writing – original draft, Writing – review and editing

## DATA AVAILABILITY

All essential data supporting this article are provided in the main text or the supporting information. All plasmids created in this work can be requested through Addgene.

## ADDITIONAL FILES

The following material is available online.

## Supplemental Material

**Supplemental Figures and Tables (Spectrum02707-23-S0001.docx).** Supplemental Tables with primers sequences, sequences of genetic parts and vector details. Supplemental figures with PCR controls of engineered strains, details of SNP analysis upon whole genome sequencing of selected strains and vector maps of the main plasmids.

## Open Peer Review

**PEER REVIEW HISTORY (review-history.pdf).** An accounting of the reviewer comments and feedback.

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
