## [Reviewer comments · Microbiology Spectrum]

Microbiology Spectrum

A SEVA-based, CRISPR-Cas3-assisted genome engineering approach for *Pseudomonas* with efficient vector curing

Eveline-Marie Lammens, Daniel Volke, Kaat Schroven, Marleen Voet, Alison Kerremans, Rob Lavigne, and Hanne Hendrix

Corresponding Author(s): Rob Lavigne, Katholieke Universiteit Leuven

Review Timeline:

Submission Date:	June 30, 2023
Editorial Decision:	August 7, 2023
Revision Received:	September 21, 2023
Editorial Decision:	October 15, 2023
Revision Received:	October 17, 2023
Accepted:	October 18, 2023

Editor: Silvia Cardona

Reviewer(s): Disclosure of reviewer identity is with reference to reviewer comments included in decision letter(s). The following individuals involved in review of your submission have agreed to reveal their identity: Christopher R Reisch (Reviewer #2)

Transaction Report:

DOI: <https://doi.org/10.1128/spectrum.02707-23>

August 7, 2023

Dr. Rob Lavigne
Katholieke Universiteit Leuven
Leuven
Belgium

Re: Spectrum02707-23 (A SEVA-based, CRISPR-Cas3-assisted genome engineering approach for *Pseudomonas* with efficient vector curing)

Dear Dr. Rob Lavigne:

Thank you for submitting your manuscript to Microbiology Spectrum. Three experts in the field evaluated your work. The reviewers noted the quality of your work but raised points that need to be addressed before the manuscript can be considered for publication. In addition, I suggest having a statement of plasmids availability (deposit to organizations like Addgene would increase the impact of the development tool).

Link Not Available

Sincerely,

Silvia Cardona

Journals Department
Reviewer comments:

Reviewer #1 (Comments for the Author):

The authors present a new CRISPR-cas approach for creating insertions, deletions, and other genetic changes in *P. putida* and *P. aeruginosa*. Researchers working with bacteria in the genus *Pseudomonas* are well aware of the challenges these bacteria present when we attempt to generate defined mutations. Some isolates are barely amenable to manipulation, and others take time and the failure rate can be high. Thus, there is always a desire and a need for improvements in the genetic tool kit for this

genus. The work presented here is therefore useful and impactful.

I think the overall process created by the research team is nicely described at the start of the results/discussion section and with figure 1. In general, the remaining text is quite clear although there are a few things I think could be clarified, outlined below.

I have no major issues with the paper and my comments are quite minor.

Line 78. Perhaps very briefly outline why the Cas9 and 12a systems don't allow genomic integration in *P. putida*.

Line 117. I would replace "his or her" with "their."

Line 157-160. I think what is written here is clear, but I don't think it matches well with the illustration in Fig 2c which shows the homology arms and looks more like a schematic for creating the HDR plasmid, rather than cloning of the spacer into the pCAS3 vector.

Line 178-180. Although briefly explained earlier when discussing selection of the PAM, I think the removal of selected recognition sites mentioned here should be explained a little more thoroughly, or a figure included to clarify. For example, figure 2 C shows the PAM in the upstream homology arm. Should this be removed? This could be used to illustrate the concept.

Line 196. What is the rationale for pooling the co-transformants rather than using individual colonies?

Line 270. Does "no substantial deletions or insertions" mean that there were not any deletions or insertions, or there were some but they were not considered "substantial"? If there were no off-target effects, I suggest say that directly with unambiguous wording. If there were some off-target changes, these should be described and some estimate of whether the type and/or frequency of these differs from what would be expected by random chance - i.e. were they generated by the CRISPR approach?

Line 313-316. If the copy number explanation is correct, it would be expected in the *P. putida* system as well, but I don't think that was the case.

Supp Info. Generally this is very useful. It might be helpful to add plasmid maps showing relevant features for the key vectors such as pCas3cRh

Reviewer #2 (Comments for the Author):

The manuscript by Lammens et al. describes a CRISPR assisted genome engineering system that functions with high efficiency in *Pseudomonas*. The system is built upon the Cas3 described by Balint et al. and moves the system to a two-plasmids built with the SEVA architecture standards. In addition a third plasmid can be used to then cure all of the plasmids. I appreciate the potential of portability and standard construction techniques.

An obvious improvement of this system would be ability to deliver repair template as linear product obtained from SOEing or Gibson assembly and amplification, without the need to clone the product first. Was this attempted? If it was attempted but unsuccessful or very low efficiency it would be informative and should be included.

I'm surprised the length of homology required for recombination was not probed. The possibility that arms could be short enough to fit on an oligo would be a boost to the system. Though I also acknowledge that clones of arms roughly 500 bp should be efficient and routine nowadays.

Certainly, the manuscript linked here is relevant and should be cited as prior art - <https://doi.org/10.1016/j.engmic.2022.100046>

References 31 and 58 are the same.

Reviewer #3 (Comments for the Author):

This paper describes the use of a CRISPR/Cas3 methodology to easily do genome engineering (deletions or insertion of heterologous DNA) in *Pseudomonas putida* and *aeruginosa*. This MS deals with an interesting topic, is well structured and well written. However, it would great if authors provide more methodological details.

However, below are some comments and questions that need to be addressed before publication.

It would help reviewers a lot if page numbers were included in the MS

Please, clearly indicate the differences of the system described in this MS versus <https://doi.org/10.1038/s41592-020-00980-w>. Please, state what is the new aspect of your work compared with DOI 10.1186/s12934-017-0748-z). It appears that the only new aspect is the combination with the curing system.

Ensure your method is compared with the Cas9 system and advantages and disadvantages are highlighted.

P2, L31. Four days appears a bit misleading. First, authors do not include the construction of the HDR and Cas3 plasmids and then also do not take into account the introduction of those plasmids into the proper strain (Fig. 1).

Please adjust the timing to a proper and more accurate situation!

P4, L118. Please, clarify the following points in the MS: What is the ori of plasmid Cas3? What is the size of the Cas3 plasmid? is it compatible with RK2 and pBBR1? Is it compatible with other SEVA oris?

Does the pCas3cRh impose a metabolic burden to cells?

Figure 1 caption, L134. Please, clarify whether the pSEVA52-oriT contains only a spacer or the CRISPR array plus sequence

P6, L147-148. These lines require more rationale. It would be great to provide more details about PAM selection (AAG vs TTC). In that line, add the 3' as well at the end of the PAM sequence and include that in the TTC seq.

P6, L155. How did the authors test that? Using Blastn? What specific conditions?

Figure 2A may not add any useful info to justify its presence. Consider its removal.

In Fig. 2C add the length of the homology arms

P7, L179-182. Please, explain that paragraph in more detail

P7, L185. Provide frequencies of co-transformation of both plasmids

P7, L186. If the efficiency of co-electroporation is insufficient, how would that affect the protocol timing? would this issue make the system similar to the I-SceI based (timewise)? connect with the statement in P11 L239.

P7, L197. After % add w/v. Apply that throughout the MS.

P8, L206-208. the reason of your statement is that recombineering genes are required to protect either ssDNA (oligo recombineering) or linear dsDNA (PCR fragments) and not plasmid-based recombination. Please, correct that sentence in the MS. In that sense, a fairer comparison of this system would have been with CRISPR/Cas9 instead of HR.

Please, explain why genome integration with two putida strains and not with *P. aeruginosa* were tested. Then, authors tested deletions in just *P. aeruginosa* and not in putida.

P9, L270. Please, explain the meaning of "no substantial deletions" in the manuscript

P9, L272. Are those two-point mutations in the integrated DNA as well? If so, it may be something to take into account with this system. Please, clarify in the manuscript.

Did authors sequenced the mutants obtained through HR? specify in the manuscript

Figure 4. Did authors test bigger DNA deletions?

P14, L370. Linearized?

P16, L429. What volume? 10 ml, 20 ml, 50 ml...?

P16, L434. What dilutions did authors plate?

Reference section. Bacterial names must be italicized L493, 496, 500, 507, etc.

Staff Comments:

Preparing Revision Guidelines

Please return the manuscript within 60 days; if you cannot complete the modification within this time period, please contact me. If you do not wish to modify the manuscript and prefer to submit it to another journal, please notify me of your decision immediately so that the manuscript may be formally withdrawn from consideration by Microbiology Spectrum.

Reviewer comments:

Reviewer #1 (Comments for the Author):

*The authors present a new CRISPR-cas approach for creating insertions, deletions, and other genetic changes in *P. putida* and *P. aeruginosa*. Researchers working with bacteria in the genus *Pseudomonas* are well aware of the challenges these bacteria present when we attempt to generate defined mutations. Some isolates are barely amenable to manipulation, and others take time and the failure rate can be high. Thus, there is always a desire and a need for improvements in the genetic tool kit for this genus. The work presented here is therefore useful and impactful.*

I think the overall process created by the research team is nicely described at the start of the results/discussion section and with figure 1. In general, the remaining text is quite clear although there are a few things I think could be clarified, outlined below.

I have no major issues with the paper and my comments are quite minor.

We thank the reviewer for her/his positive appraisal of our work.

*Line 78. Perhaps very briefly outline why the Cas9 and 12a systems don't allow genomic integration in *P. putida*.*

We have clarified this in L78.

Line 117. I would replace "his or her" with "their."

We made the suggested replacement in L117 (now L121).

Line 157-160. I think what is written here is clear, but I don't think it matches well with the illustration in Fig 2c which shows the homology arms and looks more like a schematic for creating the HDR plasmid, rather than cloning of the spacer into the pCAS3 vector.

We replaced the cross reference to Fig 2c (now labelled Fig 2b) so that it matches the explanation.

Line 178-180. Although briefly explained earlier when discussing selection of the PAM, I think the removal of selected recognition sites mentioned here should be explained a little more thoroughly, or a figure included to clarify. For example, figure 2 C shows the PAM in the upstream homology arm. Should this be removed? This could be used to illustrate the concept.

In this particular example, the protospacer/PAM sequence is interrupted upon integration of our 800 bp cassette and will no longer be cleaved by the Cas3 enzyme. Therefore, in this case, removal of the PAM site from the template is not necessary. To clarify this in the manuscript, we elaborated on this in the figure legend Fig 2c (now labelled Fig 2b) and in L165-L170.

Line 196. What is the rationale for pooling the co-transformants rather than using individual colonies?

There are two reasons for pooling the co-transformants. 1) The overnight induction with rhamnose was performed in a volume of 20 mL. Inoculating this volume from a single colony wouldn't result in sufficient cell growth overnight. 2) In the standard protocol as described in the Method section, no PCR or other validation step is performed on the co-transformants before induction of the Cas3 enzyme complex. To avoid the small risk of picking up a single co-transformant that happens to be a false-positive and would lead to negative results later in the protocol, we decided to reduce this risk significantly by picking up multiple colonies.

Line 270. Does "no substantial deletions or insertions" mean that there were not any deletions or insertions, or there were some but they were not considered "substantial"? If there were no off-target effects, I suggest say that directly with unambiguous wording. If there were some off-target changes, these should be described and some estimate of whether the type and/or frequency of these differs from what would be expected by random chance - i.e. were they generated by the CRISPR approach?

We thank the reviewer for indicating this ambiguity. We did not detect any deletions or insertions besides those listed in Tables S5-S8 and tried to phrase this (too) carefully. Therefore, we rephrased the sentence, now saying 'no deletions or insertions were detected, except for...'.

Line 313-316. If the copy number explanation is correct, it would be expected in the P. putida system as well, but I don't think that was the case.

We do detect similar trends in the *P. putida* results (Figure 3b) and the *P. aeruginosa* results (Figure 4c). For both species, the HDR vector has a lower curing efficiency than the other helper vectors for all samples. To allow better comparison of the data, we reorganized the legend of Figure 3b.

Supp Info. Generally this is very useful. It might be helpful to add plasmid maps showing relevant features for the key vectors such as pCas3cRh

Thank you for the suggestion, we have added plasmid maps of pCas3cRh, pSEVA23-PP_5388 and pSEVA52-OriT in Figure S7.

Reviewer #2 (Comments for the Author):

The manuscript by Lammens et al. describes a CRISPR assisted genome engineering system that functions with high efficiency in Pseudomonas. The system is built upon the Cas3 described by Balint et al. and moves the system to a two-plasmids built with the SEVA architecture standards. In addition a third plasmid can be used to then cure all of the plasmids. I appreciate the potential of portability and standard construction techniques.

We thank the reviewer for her/his positive evaluation of our work.

An obvious improvement of this system would be ability to deliver repair template as linear product obtained from SOEing or Gibson assembly and amplification, without the need to clone the product first. Was this attempted? If it was attempted but unsuccessful or very low efficiency it would be informative and should be included.

We thank the reviewer for this interesting suggestion. We have not (yet) attempted to introduce the repair template as a linear product. While this would remove the need for constructing the HDR vector, we believe it wouldn't necessarily save time in the protocol. For one, the pCas3cRh-spacer vector must be constructed anyway, which is done in parallel with the construction of the HDR vector. Second, the electroporation of linear products is often much less efficient than electroporation of plasmids. As such, the co-electroporation efficiency of the linear product and pCas3cRh-spacer vector might be too low to result in co-transformants, thus requiring consecutive electroporation steps and adding an extra day to the protocol. In addition, linear DNA is readily degraded in *P. putida* if no protective protein is expressed (SSR, lambda red).

I'm surprised the length of homology required for recombination was not probed. The possibility that arms could be short enough to fit on an oligo would be a boost to the system. Though I also acknowledge that clones of arms roughly 500 bp should be efficient and routine nowadays.

We selected homology arms of 500-600 bp similar to the work of Csörgő *et al.* (2020), who successfully used the pCas3cRh vector for *Pseudomonas* genome targeting. Since this also worked fine for us, we did not investigate the effect of arm length on recombination efficiency. Nevertheless, since the submission we already successfully deleted the *pilZ* gene from the *P. aeruginosa* genome using arms of around 280 bp. We added a sentence to include this information (line 160-161). For difficult to clone fragments, we suggest to order a gene block of the HDR template flanked by BsaI restriction sites. This note is added to the materials and methods section, line 357-361.

Certainly, the manuscript linked here is relevant and should be cited as prior art - <https://doi.org/10.1016/j.engmic.2022.100046>

We thank the reviewer for this suggestion. We included the reference in the introduction section as prior art for CRISPR-Cas3-based genome engineering (line 85-87).

References 31 and 58 are the same.

Thank you for bringing this error to our attention. The references of the supplementary information our now moved to the supplementary document.

Reviewer #3 (Comments for the Author):

*This paper describes the use of a CRISPR/Cas3 methodology to easily do genome engineering (deletions or insertion of heterologous DNA) in *Pseudomonas putida* and *aeruginosa*. This MS deals with an interesting topic, is well structured and well written. However, it would great if authors provide more methodological details.*

Thank you for this positive assessment of our work and your suggestions to improve the manuscript with more methodological details.

However, below are some comments and questions that need to be addressed before publication.

It would help reviewers a lot if page numbers were included in the MS.

We apologize for the inconvenience and have added page numbers to the revised manuscript and supplementary information.

Please, clearly indicate the differences of the system described in this MS versus <https://doi.org/10.1038/s41592-020-00980-w>. Please, state what is the new aspect of

your work compared with DOI 10.1186/s12934-017-0748-z). It appears that the only new aspect is the combination with the curing system.

Although we used the targeting plasmid of Csörgo et al., we optimized the system by (1) improving the cloning efficiencies by placing the HDR template on a second plasmid instead of on the relatively large targeting plasmid, (2) providing a vector set of both plasmids with multiple antibiotic markers for application in various Gram-negative hosts and different designs, and (3) applying the CRISPR-Cas3 system for downstream curing. Initially, we tried to include the HDR on the pCas3cRh plasmid, as was done by Csörgo et al. However, low cloning and transformation efficiencies were obtained. The low efficiencies might be the result of the large vector size (> 10 kbp). Therefore, we switched to a two-plasmid approach with universal SEVA plasmids, making it more easy to use. We added the extra information to the conclusion section (line 305-309).

As mentioned in line 225, a similar concept of vector curing using CRISPR-Cas3 instead of CRISPR-Cas9 is used. It is the combination of using a plasmid with CRISPR-Cas3 system for both engineering and curing purpose that make our concept novel. Moreover, we only introduced one spacer targeting a specific sequence in the oriT region (present in all plasmids of our system), while Lauritsen et al. introduced multiple spacers targeting different conserved plasmid regions (e.g. oriR) as they aimed to cure also naturally-occurring plasmids. Our system can be adapted to also cure more general plasmids (synthetic and naturally-occurring) by introducing multiple spacers, as explained in the text (line 321 and 322).

Ensure your method is compared with the Cas9 system and advantages and disadvantages are highlighted.

We added some extra information in the introduction section, highlighting the combined nuclease-helicase of Cas3 (line 81-82) and the previously observed lower engineering efficiencies in *Pseudomonas* using Cas9 (77-79 and 94-96). In addition, throughout the text comparisons in efficiencies of the Cas3-assisted system over the Cas9-assisted systems are cited (line 194-196, 247-249, 313-314).

P2, L31. Four days appears a bit misleading. First, authors do not include the construction of the HDR and Cas3 plasmids and then also do not take into account the introduction of those plasmids into the proper strain (Fig. 1).

Please adjust the timing to a proper and more accurate situation!

In our hands, the actual genomic engineering starting from the electroporation of the proper strain to the PCR screening of integration/deletion mutants is standardly performed in four days. To include some time for error, we have edited L31 from 'four days' to 'a week'.

Furthermore, we did include the introduction of plasmids in *Pseudomonas* into our 4-day calculation, as depicted in Fig 1 ("Day 1: The pCas3cRh vector with spacer sequence (red) for the target site and pSEVA231 with the repair template for homology-directed repair (pSX3-HDR) are introduced in *Pseudomonas* by means of electroporation.").

P4, L118. Please, clarify the following points in the MS: What is the ori of plasmid Cas3? What is the size of the Cas3 plasmid? is it compatible with RK2 and pBBR1? Is it compatible with other SEVA oris? Does the pCas3cRh impose a metabolic burden to cells?

The pCas3cRh vector carries the *pRO1600/ColE1* ori (as mentioned in Table S2), which is known to be compatible with the R6K, RK2, pBBR1 and RSF1010 origins of the SEVA collection. For clarity, we have added L123-124 "A notable exception are vectors carrying the pRO1600/ColE1 origin of replication, as this is the replicative origin of the pCas3cRh vector (Table S2)". Furthermore, vector maps of pCas3cRh, pSEVA23-PP_5388 and pSEVA52-oriT were added in the supplementary information (Figure S7).

We did observe that cells grew more slowly when expression of the Cas3 enzymes was induced from pCas3cRh, which was also mentioned in L267-268. However, as this vector is cured from the cells after the genomic edit is made and no noteworthy off-target mutations were detected in the resulting mutants, we believe that the metabolic burden of this vector should not be considered as troublesome.

Figure 1 caption, L134. Please, clarify whether the pSEVA52-oriT contains only a spacer or the CRISPR array plus sequence

Thank you for indicating that this is not clear to the reader. The vector contains a CRISPR array with a spacer targeting oriT. To clarify this, we have 1) mentioned this in the Figure legend L634 and, 2) added the vector map of this vector in the supplementary information (Figure S7).

P6, L147-148. These lines require more rationale. It would be great to provide more details about PAM selection (AAG vs TTC). In that line, add the 3' as well at the end of the PAM sequence and include that in the TTC seq.

We apologize for the miscommunication. Both the AAG and TTC PAMs are identical, but they are located on the sense or antisense strand, respectively. We have changed the phrasing in the manuscript to clarify this (L136-139).

P6, L155. How did the authors test that? Using Blastn? What specific conditions?

Indeed, we used Blastn to check for potential off target hits against the bacterial genome, using the standard settings of the tool (L147).

Figure 2A may not add any useful info to justify its presence. Consider its removal. In Fig. 2C add the length of the homology arms

We have removed 2A and added the length of the homology arms in 2C (now 2B) as requested.

P7, L179-182. Please, explain that paragraph in more detail

We provided additional information on the PAM/protospacer interruption in this paragraph and added a vector map of pSEVA23-PP_5388 in Figure S7. Details on the vector construction are already described in the Methods section and all used primers and sequences of genetic parts are available in the supplementary information.

P7, L185. Provide frequencies of co-transformation of both plasmids

We provided to transformation efficiencies for the co-transformation of both *P. putida* strains in L176.

P7, L186. If the efficiency of co-electroporation is insufficient, how would that affect the protocol timing? would this issue make the system similar to the I-SceI based (timewise)? connect with the statement in P11 L239.

For *P. putida*, this would add one extra day to the protocol (added in L174), while for *P. aeruginosa* this would not make a difference in timing as co-electroporation requires a 2-day incubation time, whereas separate electroporation steps should allow overnight colony formation.

Despite the additional day for electroporation, this protocol would still be shorter than the I-SceI method as it does not require multiple restreaking steps to avoid colonies with mixed-phenotypes. Even if it would take the same amount of time, the Cas3 based method shows engineering efficiencies that significantly outperform the I-SceI-based homologous recombination approach.

P7, L197. After % add w/v. Apply that throughout the MS.

We added (w/v) throughout the MS, whenever applicable.

P8, L206-208. the reason of your statement is that recombineering genes are required to protect either ssDNA (oligo recombineering) or linear dsDNA (PCR fragments) and not plasmid-based recombination. Please, correct that sentence in the MS. In that sense, a fairer comparison of this system would have been with CRISPR/Cas9 instead of HR.

Genomic integration in *P. putida* is not possible with the sole use of the CRISPR-Cas9 system. When performing integrations, the CRISPR-Cas9 system can be used in combination

with standard homologous recombination or recombineering approaches. This is in contrast to the CRISPR-Cas3 system which can be used as a stand-alone method for genomic integrations in *P. putida*. We have added more information of the limitations of the Cas9 system in the introduction (L78-79) and adjusted the statement in L206-208 (now L195-196) to make this more clear in the manuscript.

Please, explain why genome integration with two putida strains and not with P. aeruginosa were tested. Then, authors tested deletions in just P. aeruginosa and not in putida.

We aimed to give a general overview of the potential of our system by showing specific examples, selecting two different species (and two different strains) and showing an example of integration and deletions. Nowadays, more than ten *Pseudomonas* strains with an integration, deletion or substitution (or combinations) are already successfully constructed, demonstrating the wide use of this protocol.

P9, L270. Please, explain the meaning of "no substantial deletions" in the manuscript

We thank the reviewer for indicating this ambiguity. We did not detect any deletions or insertions besides those listed in Tables S5-S8 and tried to phrase this (too) carefully. Therefore, we rephrased the sentence, now saying 'no deletions or insertions were detected, except for...'.

P9, L272. Are those two-point mutations in the integrated DNA as well? If so, it may be something to take into account with this system. Please, clarify in the manuscript.

None of the observed mutations were located within the integration cassette. To highlight this, we have added an additional sentence in L290 and mentioned the nucleotide position of the integration cassette in the table legends of Tables S5-S8.

Did authors sequenced the mutants obtained through HR? specify in the manuscript

The mutants that were created through HR were not cured of the helper vectors and therefore, we did not sequence them. They only served as a comparison for the engineering efficiency of the novel Cas3-assisted engineering method.

Figure 4. Did authors test bigger DNA deletions?

Yes, we successfully deleted the prophage D3 from the *P. aeruginosa* PAO1 genome. This information is added to the conclusion section (line 315-318).

P14, L370. Linearized?

We rephrased linearized to PCR-amplified pSEVAX3 backbone to make this more clear. The PCR amplification is described in the beginning of the same paragraph.

P16, L429. What volume? 10 ml, 20 ml, 50 ml...?

We clarified the methodology for preparing electrocompetent cells by specifying the culture volume and washing volume (line 405).

P16, L434. What dilutions did authors plate?

We clarified the methodology (line 411).

Reference section. Bacterial names must be italicized L493, 496, 500, 507, etc.

All bacterial names should now be italicized.

October 15, 2023

Dr. Rob Lavigne
Katholieke Universiteit Leuven
Leuven
Belgium

Re: Spectrum02707-23R1 (A SEVA-based, CRISPR-Cas3-assisted genome engineering approach for *Pseudomonas* with efficient vector curing)

Dear Dr. Rob Lavigne:

Thank you for submitting your manuscript to Microbiology Spectrum. As you will see your paper is very close to acceptance. Please modify the manuscript along the lines the reviewer has recommended. I agree with the reviewer that lines 78-79 are a bit confusing. I recommend going to line 74 and ensuring the reader knows whether the paragraph refers to CRISPR with a template provided for HR or CRISPR used as a counterselection method in *P. putida*. Please ensure "NHES" is defined. Regarding the specific reviewer's comment, you can explain why the endogenous non-homologous end repair system can affect the targeted modification or simply remove the suggestion and just state that the method have low efficiency.

As these revisions are quite minor, I expect that you should be able to turn in the revised paper in less than 30 days, if not sooner. If your manuscript was reviewed, you will find the reviewers' comments below.

When submitting the revised version of your paper, please provide (1) point-by-point responses to the issues raised by the reviewers as file type "Response to Reviewers," not in your cover letter, and (2) a PDF file that indicates the changes from the original submission (by highlighting or underlining the changes) as file type "Marked Up Manuscript - For Review Only". Please use this link to submit your revised manuscript. Detailed instructions on submitting your revised paper are below.

Link Not Available

Sincerely,

Silvia Cardona

Reviewer comments:

Reviewer #3 (Comments for the Author):

This is a revised MS in which authors described a method based on the use of a CRISPR/Cas3 methodology to easily do genome engineering (deletions or insertion of heterologous DNA) in *Pseudomonas putida* and *aeruginosa*.

Authors have answered all questions I posed and revised the MS accordingly. That, and the suggestion of the other reviewers help to improve the MS. Now, I can only congratulate authors for the MS!

However, I would just revise the following minor thing:

I'm not convinced about the recently added comment (P3, L78-79) about the reason that does not allow genome integration of heterologous DNA in *P. putida*.

Preparing Revision Guidelines

Please return the manuscript within 60 days; if you cannot complete the modification within this time period, please contact me. If you do not wish to modify the manuscript and prefer to submit it to another journal, please notify me of your decision immediately so that the manuscript may be formally withdrawn from consideration by Microbiology Spectrum.

Reviewer comments:

Reviewer #3 (Comments for the Author):

This is a revised MS in which authors described a method based on the use of a CRISPR/Cas3 methodology to easily do genome engineering (deletions or insertion of heterologous DNA) in *Pseudomonas putida* and *aeruginosa*.

Authors have answered all questions I posed and revised the MS accordingly. That, and the suggestion of the other reviewers help to improve the MS. Now, I can only congratulate authors for the MS!

We thank the reviewer for their positive appraisal of our work!

However, I would just revise the following minor thing:

I'm not convinced about the recently added comment (P3, L78-79) about the reason that does not allow genome integration of heterologous DNA in *P. putida*.

*Upon request of the editor, we clarified the statement in L78-79. The exact reason for the very low integration efficiencies of CRISPR-Cas9 in *P. putida* has not been studied to our knowledge. However, in our opinion the competition with NHEJ is the most plausible explanation and we did try to phrase this as such by using 'probable' in L80.*

October 18, 2023

Dr. Rob Lavigne
Katholieke Universiteit Leuven
Leuven
Belgium

Re: Spectrum02707-23R2 (A SEVA-based, CRISPR-Cas3-assisted genome engineering approach for *Pseudomonas* with efficient vector curing)

Dear Dr. Rob Lavigne:

Your manuscript has been accepted, and I am forwarding it to the ASM Journals Department for publication. You will be notified when your proofs are ready to be viewed.

Sincerely,

Silvia Cardona
Editor, Microbiology Spectrum
